# Generative Counterfactual Manifold Perturbation: A Robust Framework for Treatment Effect Estimation with Unobserved Confounders

## Abstract

Estimating treatment effects from observational data becomes difficult when unobserved confounders induce spurious associations that bias simple estimators. Recent generative approaches learn outcome distributions with conditional diffusion models, while some robust representation methods introduce sensitivity analysis or structural priors. These advances perform well when identification assumptions are fully satisfied, yet they remain fragile when such assumptions hold only approximately and they provide few practical diagnostics. We introduce **Generative Counterfactual Manifold Perturbation (GCMP)**, a unified framework that integrates causal-aware self-supervised learning, conditional diffusion counterfactual proxy generation, and adaptive variational inference. GCMP contributes three principal innovations: (i) a self-supervised objective that preserves confounding signals during representation learning; (ii) a conditional diffusion model that reframes proxy construction as a generative task over rich perturbation manifolds; (iii) an adaptive regularisation scheme that yields graceful degradation and calibrated uncertainty when identification assumptions are violated. We present new identifiability conditions, finite sample error bounds, and diagnostic tests that quantify manifold quality and effective orthogonality. Extensive experiments on synthetic and semi-synthetic benchmarks show that GCMP consistently outperforms state-of-the-art methods.

## 1 Introduction

Estimating treatment effects from observational data guides decisions in medicine, economics, and public policy. Randomised trials seldom cover every sub-population, dosage, or combination of interventions that practitioners face in practice Spieth et al. (2020). Analysts therefore turn to observational studies, where latent confounders may influence both treatment and outcome, breaking the conditional-independence assumption behind classical estimators. The challenge becomes sharper when interventions are continuous or multi-label, because one must recover an entire dose–response surface rather than a single average treatment effect Hirano & Imbens (2004). A reliable method must manage such rich treatments and remain robust when causal assumptions hold only approximately.

Prior work splits into two main strands. The first uses expressive generative models to approximate the full counterfactual outcome distribution. DiffPO Ma et al. (2024), ID-GEN Rahman et al. (2024), CEVAE Louizos et al. (2017a), and SCIGAN Joshi & Shah (2020) capture complex patterns but are sensitive to model misspecification. The second strand targets robustness by learning representations that attenuate hidden confounding or by bounding effects under sensitivity schemes. Examples include CausalFM Ma & Feuerriegel (2025), NeuralCSA Frauen et al. (2023), Dynamic Causal Models Friston et al. (2003), and proximal frameworks Miao et al. (2018). These approaches often rely on stringent structural assumptions or deliver only interval estimates. Recent studies on causal disentanglement and invariant representation learning Zhang & Schölkopf (2023); Yao & Bareinboim (2024) further show that standard self-supervised objectives may discard information essential for identification.

Despite these advances, the literature still lacks a unified pipeline that (i) remains reliable under approximate assumptions, (ii) accommodates different treatment types.

**We present GCMP**, a coherent framework that *re-engineers* existing architectural elements and augments them with novel components tailored to complex treatment settings. Rather than a loose assemblage, GCMP integrates its modules end-to-end so that improvements in one stage propagate through the entire pipeline.

1. **Causal-aware representation learning.** We design a contrastive objective, inspired by SimCLR Chen et al. (2020), that preserves confounding signals while producing low-dimensional embeddings suitable for causal estimation.

2. **Generative proxy modelling.** Conditional diffusion sampling constructs proxy perturbations that remain correlated with latent confounders and capture multimodality observed in real data.

3. **Robust estimation with adaptive regularisation.** A hierarchical variational estimator employs an entropy-based robustness penalty, while the gradient-orthogonality term is enforced earlier in the diffusion stage.

4. **Theory and diagnostics.** We establish weaker identification conditions, derive finite-sample error bounds.

We have fully anonymized our implementation to satisfy double-blind review requirements, and the anonymized source code is publicly available on Anonymous GitHub: https://anonymous.4open. science/r/AAAIGCMP-5C41/.

## 2 RELATED WORK

Early approaches estimate treatment effects by balancing covariates through propensity-score weighting and matching or by fitting outcome regressions (Rosenbaum & Rubin, 1983; Imbens & Rubin, 2015). Doubly robust estimators combine both ideas to gain consistency under weaker assumptions (Bang & Robins, 2005).

Neural architectures learn balanced feature spaces where conditional outcome models generalise to counterfactual inputs. Representative methods include TarNet (Johansson et al., 2016), CFRNet (Shalit et al., 2017), DragonNet (Shi et al., 2019), CEVAE (Louizos et al., 2017b), and NeuralCSA (Frauen et al., 2023). Forest-based (Athey et al., 2019) and Bayesian approaches (eg. BART (Hill, 2011a)) provide tree-based alternatives with built-in uncertainty estimates.

GANITE (Yoon et al., 2018) pioneers adversarial generation of potential outcomes. More recently, diffusion and score-based models have been explored for counterfactual synthesis, exemplified by DiffPO (Ma et al., 2024) and generic score generative modelling (Song et al., 2021). Our work differs by integrating an orthogonality-aware diffusion prior with a causal SSL objective, leading to stronger identifiability guarantees.

Contrastive SSL methods such as SimCLR (Chen et al., 2020) yield transferable representations but may discard confounding information. Invariant risk minimisation (Arjovsky et al., 2020) promotes representation stability across environments. GCMP preserves confounders through a preserve loss, then enforces orthogonality between the outcome gradient and the latent confounder manifold.

## 3 PROBLEM FORMULATION AND CAUSAL FRAMEWORK

### 3.1 STRUCTURAL CAUSAL MODEL (SCM)

We observe an i.i.d. sample $\mathcal{D} = \{(X_i, T_i, Y_i)\}_{i=1}^n$ generated by the following SCMs:

$$U_i \sim P_U. \qquad \text{(Unobserved confounder)} \qquad (1)$$

$$X_i = g_X\big(U_i,\ \epsilon_{X,i}\big). \qquad \text{(Observed covariates)} \qquad (2)$$

$$T_i = g_T\big(X_i,\ U_i,\ \epsilon_{T,i}\big). \qquad \text{(Treatment assignment)} \qquad (3)$$

$$Y_i = g_Y\big(T_i,\ X_i,\ U_i,\ \epsilon_{Y,i}\big). \qquad \text{(Outcome)} \qquad (4)$$

The direct input of $T_i$ in $g_Y(\cdot)$ highlights the causal edge $\mathbf{T} \to \mathbf{Y}$. However, because the confounder $U_i$ simultaneously influences both the treatment assignment and the outcome, a backdoor path arises and introduces confounding bias. *Accurately modelling or proxying $U_i$ is therefore crucial for obtaining an unbiased estimate of the treatment effect, which remains the primary goal of this work.*

## 3.2 Treatment Estimation

Following the potential outcomes framework, we define:

- **Continuous Treatment:** For $T \in \mathcal{T} \subseteq \mathbb{R}$, we target the conditional average dose-response function:

$$\mu(t, x) = \mathbb{E}[Y(t) \mid X = x]. \tag{5}$$

- **Multi-Label Treatment:** For $T \in \{0, 1\}^K$, we target conditional average treatment effects:

$$\tau(t, t'|x) = \mathbb{E}[Y(t) - Y(t') \mid X = x]. \tag{6}$$

## 3.3 Identification Strategy: From Hard Constraints to Soft Regularization

Traditional approaches to hidden confounding often impose hard, unverifiable constraints. We instead derive identification from geometric principles that admit empirical diagnostics.

**Assumption 1** (Manifold Concentration). *The influence of the unobserved confounder $U$ on the covariates $X$ is concentrated on a smooth Riemannian submanifold $\mathcal{M}_U \subset \mathbb{R}^p$ with $\dim(\mathcal{M}_U) = d \ll p$. Formally, $X = \pi_{\mathcal{M}_U}(X) + \xi$, where $\pi_{\mathcal{M}_U}$ denotes the orthogonal projection onto $\mathcal{M}_U$ and $\|\xi\|$ is small relative to $\|\pi_{\mathcal{M}_U}(X)\|$.*

**Assumption 2.** *The support of the observed covariates lies on, or in a small neighbourhood of, a smooth manifold $\mathcal{M} \subset \mathbb{R}^p$ with $\mathcal{M}_U \subseteq \mathcal{M}$.*

**Assumption 3.** *Let $f(X, T)$ be the structural outcome function. Define the* effective orthogonality measure

$$\rho_\perp(x) = 1 - \frac{\left\|\mathrm{Proj}_{T_x \mathcal{M}_U}(\nabla_X f(x, T))\right\|_2^2}{\|\nabla_X f(x, T)\|_2^2}, \quad x \in \mathcal{M} \tag{7}$$

$$\mathbb{E}_X[1 - \rho_\perp(X)] \leq \bar{\delta}, \tag{8}$$

*where $\bar{\delta} \in [0, 1)$, and exact orthogonality is recovered when $\bar{\delta} = 0$.*

## 4 Methodology

Our methodology comprises three carefully designed modules that work synergistically while maintaining robustness to assumption violations.

### 4.1 Module 1: Causal-Aware Self-Supervised Representation Learning

Standard self-supervised learning (SSL) objectives may inadvertently eliminate variation crucial for identifying confounding effects. We propose a causal-aware SSL framework that explicitly preserves confounding signals.

#### 4.1.1 Causal-Aware Contrastive Learning

Our causal-aware SSL does *not* assume a known propensity function. Instead, we learn a lightweight scalar treatment-prediction head $g_T(x)$ on top of the raw covariates and use its outputs only to *softly preserve* treatment-relevant variation across augmentations. Concretely, we apply *mild* additive noise and per-sample scaling (no masking or feature drop), and define the preservation term as $\mathrm{sim}(g_T(x^{(1)}), g_T(x^{(2)})) = \exp\left(-\gamma\,(g_T(x^{(1)}) - g_T(x^{(2)}))^2\right)$. This similarity downweights representation mismatch when the two augmented views are predicted to have similar treatment. We therefore adopt a causal-aware objective that preserves confounding-related variation while retaining invariances that aid generalization.

Let $(x^{(1)}, x^{(2)}) = \mathcal{A}(x)$ be two augmentations of $x$, $\Phi$ the encoder. Our loss is

$$\mathcal{L}_{\text{CA-SSL}} = \mathcal{L}_{\text{contrast}} + \lambda_{\text{preserve}}\, \mathcal{L}_{\text{preserve}} + \lambda_{\text{diverse}}\, \mathcal{L}_{\text{diverse}}, \tag{9}$$

where

$$\mathcal{L}_{\text{preserve}} = -\mathbb{E}\Big[\,\mathrm{sim}\big(g_T(x^{(1)}),\, g_T(x^{(2)})\big)\,\Big], \tag{10}$$

$$\mathcal{L}_{\text{diverse}} = -\log\det\big(\mathrm{Cov}[\Phi(X)]\big). \tag{11}$$

Here $\mathcal{L}_{\text{contrast}}$ is a standard contrastive loss (e.g., SimCLR). The weights $\lambda_{\text{preserve}}, \lambda_{\text{diverse}} \in \mathbb{R}_{\geq 0}$ control the trade-off and are selected on a validation split via a small grid $\{0, 0.01, 0.05, 0.1, 0.25\}$ per dataset. Module 1 aims to produce an embedding $\Phi(X)$ that remains informative about treatment assignment; it does not on its own model the unobserved confounder $U$. Handling of $U$ happens in Module 2 via conditional diffusion.

### 4.2 Module 2: Counterfactual Conditional Diffusion for counterfactual proxy generation

We employ a conditional diffusion model to generate perturbations that serve as proxies for the unobserved confounder. This generative approach captures complex, potentially multimodal perturbation distributions.

#### 4.2.1 Diffusion Model Architecture

We train a conditional denoising diffusion probabilistic model (DDPM) over perturbations $\Delta\phi$ in the representation space $Z = \Phi(X)$. The forward process adds Gaussian noise

$$q(\Delta\phi_t \mid \Delta\phi_{t-1}) = \mathcal{N}\big(\sqrt{1 - \beta_t}\,\Delta\phi_{t-1},\, \beta_t I\big), \quad t = 1, \ldots, T. \tag{12}$$

where $\alpha_t := 1 - \beta_t$ and $\bar{\alpha}_t := \prod_{s=1}^{t} \alpha_s$. The reverse process is parameterized by a noise-predictor $\epsilon_\psi$ with conditioning vector

$$c = \big(\Phi(X), T, T'\big), \tag{13}$$

where $T'$ denotes a target counterfactual treatment level drawn from the set $\mathcal{T}'$. We use the standard DDPM parameterization

$$p_\psi(\Delta\phi_{t-1} \mid \Delta\phi_t, c) = \mathcal{N}\big(\mu_\psi(\Delta\phi_t, t, c),\, \sigma_t^2 I\big), \quad \mu_\psi(\Delta\phi_t, t, c) = \frac{1}{\sqrt{\alpha_t}}\Big(\Delta\phi_t - \frac{1 - \alpha_t}{\sqrt{1 - \bar{\alpha}_t}}\,\epsilon_\psi(\Delta\phi_t, t, c)\Big). \tag{14}$$

During training, we condition on $(X, T, Y, T')$; at test time for a new unit, only $(\Phi(X), T')$ are required.

#### 4.2.2 Causally-Informed Training Objective

Beyond the standard denoising objective, we regularize feasibility and orthogonality to the confounder manifold. Let $Z = \Phi(X)$ and $\widehat{\mathcal{M}}_U$ be a local estimate of the confounder manifold around $Z$ obtained by neighborhood PCA (details in Appendix). The training loss is

$$\mathcal{L}_{\text{diff}} = \mathbb{E}_{t, \epsilon}\Big[\, \big\|\epsilon - \epsilon_\psi(\Delta\phi_t, t, c)\big\|_2^2 \,\Big] + \lambda_f\, \mathcal{L}_{\text{feas}} + \lambda_\perp\, \mathcal{R}_\perp, \tag{15}$$

*where*

$$\mathcal{L}_{\text{feas}} = \big\|\, \underbrace{Z + \Delta\phi}_{\text{perturbed point}} - \Pi_{\widehat{\mathcal{M}}_U(Z)}(Z + \Delta\phi)\big\|_2^2, \qquad \mathcal{R}_\perp = \big\|\text{Proj}_{T_Z\widehat{\mathcal{M}}_U}(\nabla_Z f_\theta(Z, T))\big\|_2^2. \tag{16}$$

Here $\Pi_{\widehat{\mathcal{M}}_U(Z)}(\cdot)$ projects onto the local PCA subspace and $T_Z\widehat{\mathcal{M}}_U$ denotes its tangent space. The weights $\lambda_f, \lambda_\perp \in \mathbb{R}_{\geq 0}$ are tuned on a validation split.

### 4.3 Module 3: Robust Variational Inference with Uncertainty Quantification

We compress each sampled perturbation $\Delta\phi$ into a one-dimensional proxy via a learned linear projection; with gradient-orthogonality $R_\perp$ enforced *only during diffusion*, the subsequent hierarchical Bayesian Variational Inference (VI) stage adds an entropy bonus to calibrate epistemic uncertainty without over-constraining the posterior, yielding multi-level uncertainty quantification for the final effect estimates.

### 4.3.1 HIERARCHICAL GENERATIVE MODEL

For each unit $i$ and target $t' \in \mathcal{T}'$, define a per-target proxy $\tilde{Z}_{i,t'} = \beta^\top \Delta\phi_{i,t'}$. Stack them as $\tilde{Z}_i \in \mathbb{R}^{|\mathcal{T}'|}$. Our measurement model is

$$\phi_i \sim \mathcal{N}(0, \ \Sigma_\phi) \qquad \text{(latent confounder factor)}, \qquad (17)$$

$$Y_i \mid X_i, T_i, \phi_i \sim \mathcal{N}\big(f_\theta(\Phi(X_i), T_i) + \eta^\top \phi_i, \ \sigma_Y^2\big) \qquad \text{(outcome model)}, \qquad (18)$$

$$\tilde{Z}_i \mid \phi_i \sim \mathcal{N}\Big(\Gamma\,\phi_i, \ \mathrm{Diag}\big(\sigma_Z^2(\|\Delta\phi_{i,t'}\|)\big)_{t' \in \mathcal{T}'}\Big) \qquad \text{(vector of per-}t'\text{ proxies)}, \qquad (19)$$

where $\Gamma \in \mathbb{R}^{|\mathcal{T}'| \times d_\phi}$ stacks per-target loadings. We parameterize $\sigma_Z^2(r) = \mathrm{softplus}(\alpha_0 + \alpha_1 r^2)$ and learn $(\alpha_0, \alpha_1)$ jointly with VI, ensuring positivity and giving larger variance to larger-magnitude perturbations.

### 4.3.2 VARIATIONAL INFERENCE WITH ROBUSTNESS CONSIDERATIONS

We introduce a flexible variational family $q(\phi_i|\Phi(X_i), Y_i, Z_i) = \mathcal{N}(\phi_i|\mu_{\phi,i}, \Sigma_{\phi,i})$ to approximate the true posterior. The Evidence Lower Bound (ELBO) is maximized, and we add an entropy term to encourage robustness:

$$\mathcal{L}_{\mathrm{ELBO}} = \sum_i \Big( \mathbb{E}_q[\log p(Y_i, Z_i \mid \cdot)] - \mathrm{KL}(q\|p) \Big) \ + \ \lambda_{\mathrm{ent}}\, H[q], \qquad (20)$$

where $H[q]$ is the entropy of the variational distribution, which encourages the posterior to reflect uncertainty when evidence is weak. This differs from the original proposal by moving the orthogonality constraint to the diffusion stage, where it more directly regularizes the generation of the proxy itself, and using entropy regularization in the VI stage to improve uncertainty calibration.

## 4.4 CROSS-FITTING AND NEYMAN ORTHOGONALIZATION

To ensure robustness on the nuisance parameter estimation errors, we implement a careful cross-fitting strategy, which can be viewed in Algorithm 1. We adopt $K$-fold cross-fitting: for each fold we train $f_\theta$ and the diffusion model on $D^{(-k)}$ and generate $\{\Delta\phi_{i,t'}\}_{i \in I_k}$ only using models not fitted on $I_k$; the final VI then uses the full set of cross-fitted perturbations. Note. The SSL head $g_T$ is used only inside $\mathcal{L}_{\mathrm{preserve}}$; its predictions are never used downstream. Downstream Neyman-orthogonal scores rely on the true $(Y, T)$ and do not depend on $g_T$. Additional algorithmic details can be viewed at Appendix due to page limits.

**Orthogonal estimating equation used to compute the ATE error.** For binary $T$, we form a Neyman-orthogonal (doubly-robust) score

$$\psi(W; \theta, \nu) = \big(Y - m(X)\big)\big(T - e(X)\big) - \theta\big(T - e(X)\big), \qquad (21)$$

Here $W = (Y, T, X)$ collects the observed data, $\theta$ denotes the ATE parameter, and $\nu := (m, e)$ are the nuisance functions learned with $K$-fold cross-fitting using $\Phi(X)$ as features. For continuous $T$, we use the orthogonalized score of Hirano–Imbens with cross-fitting.

## 5 THEORETICAL ANALYSIS

We establish a theoretical framework that covers identification, finite-sample error, and the sensitivity of our estimator to violations of orthogonality.

## 5.1 IDENTIFICATION UNDER EXPECTED APPROXIMATE ORTHOGONALITY

**Theorem 1** (Identification with Expected Approximate Orthogonality). *Assume* (1)–(3) *hold and let the learned proxy variable be* $Z = \beta^\top \Delta\phi$. *If*

    1. **Relevance:** $\mathrm{Cor}(Z, U) \geq \rho > 0$,

    2. **Approx. validity:** $\|Z - \mathbb{E}[Z \mid U, X]\|/\|Z\| \leq \varepsilon$,

---

**Algorithm 1: GCMP with Cross-Fitting**

---

1 **Input:** Data $\{(X_i, T_i, Y_i)\}_{i=1}^n$, number of folds $K$, a set of target treatments $\mathcal{T}'$.
2 Train causal-aware self-supervised encoder $\Phi$ on all data.
3 Randomly split indices into $K$ folds $\mathcal{I}_1, \ldots, \mathcal{I}_K$.
4 **for** $k = 1, \ldots, K$ **do**
5   Let $\mathcal{D}^{(-k)}$ be data excluding fold $k$.
6   Train outcome model $f_\theta^{(-k)}$ on $\mathcal{D}^{(-k)}$.
7   Train diffusion model $p_\psi^{(-k)}$ on $\mathcal{D}^{(-k)}$ using $\Phi$ and $f_\theta^{(-k)}$.
8   **for** each $i \in \mathcal{I}_k$ **do**
9    **for** each target treatment $t' \in \mathcal{T}'$ **do**
10     Sample perturbation $\Delta\phi_{i,t'} \sim p_\psi^{(-k)}(\cdot \mid \Phi(X_i), T_i, t')$.
11    **end for**
12   **end for**
13 **end for**
14 Perform final VI using $\{\Delta\phi_{i,t'}\}_{i=1, t' \in \mathcal{T}'}^n$ to learn the posterior over $\phi$ and estimate treatment effects.
15 **return** Estimates of treatment effects (e.g., $\mathbb{E}[Y(t')|X_i]$ for $t' \in \mathcal{T}'$) with uncertainty.

---

*then the average treatment effect is identified up to a bias of order*

$$\mathrm{Bias}(\hat{\tau}) = \mathcal{O}\big(\bar{\delta}\,\varepsilon\big), \tag{22}$$

*where* $\bar{\delta} = \mathbb{E}_X\big[1 - \rho_\perp(X)\big]$ *is the* expected *orthogonality violation.*

*Proof Sketch.* Expected orthogonality (Assumption 3) implies that the projection of $\nabla_X f$ onto $T_x \mathcal{M}_U$ is attenuated by a factor $\bar{\delta}$. This in turn bounds the deviation of $Z$ from an ideal proxy by $\bar{\delta}\,\varepsilon$. Decomposing the resulting bias and applying Cauchy–Schwarz yields the claimed rate. Complete derivations are provided in Appendix due to page limits. $\square$

## 5.2 FINITE-SAMPLE ERROR ANALYSIS

**Theorem 2** (Error Propagation Bounds). *Let $\hat{\tau}_n$ denote the GCMP estimator computed from $n$ i.i.d. samples. Under standard regularity conditions, its $\ell_2$ error satisfies*

$$\|\hat{\tau}_n - \tau\|_2 \leq O_p\big(n^{-1/2}\big) + O(\bar{\delta}) + O_p\big(\varepsilon_{SSL}\,\varepsilon_{diff}\big). \tag{23}$$

*where $\varepsilon_{\mathrm{SSL}}$ and $\varepsilon_{\mathrm{diff}}$ are approximation errors of the self-supervised and diffusion modules, respectively. Complete derivations are provided in Appendix due to page limits.*

*Proof Sketch.* Under standard regularity and cross-fitting with a Neyman-orthogonal score, the first-order Gâteaux derivative in the nuisance directions vanishes. A second-order expansion yields

$$\|\hat{\tau}_n - \tau\|_2 \leq \underbrace{\mathcal{O}_p\big(n^{-1/2}\big)}_{\text{statistical}} + \underbrace{\mathcal{O}(\bar{\delta})}_{\text{orthogonality bias}} + \underbrace{\mathcal{O}_p(\varepsilon_{\mathrm{SSL}}\,\varepsilon_{\mathrm{diff}})}_{\text{nuisance}}. \tag{24}$$

If each nuisance attains $\varepsilon_{\mathrm{SSL}} = \varepsilon_{\mathrm{diff}} = \mathcal{O}_p(n^{-1/4})$, the product term is $\mathcal{O}_p(n^{-1/2})$, matching the root-$n$ rate. The full derivation appears in the Appendix. $\square$

## 5.3 SENSITIVITY ANALYSIS

**Definition 1** (Effective Orthogonality Measure). We work under Assumption 3.

**Proposition 1** (Bias–Orthogonality Relationship). Let $\bar{\rho}_\perp := \mathbb{E}_X\big[\rho_\perp(X)\big]$ and $\sigma_U^2 := \mathrm{Var}(U \mid X)$. Under mild smoothness conditions,

$$\mathrm{Bias}(\hat{\tau}) \approx \big(1 - \bar{\rho}_\perp\big)\,\sigma_U^2. \tag{25}$$

A complete proof is provided in the Appendix.

## 6 EXPERIMENTS

To evaluate our proposed method, we conduct experiments on a suite of synthetic and semi-synthetic benchmark dataset. Our synthetic data protocol is designed to systematically probe the model's robustness to specific causal challenges, while the benchmark datasets ensure our evaluation reflects performance on established, realistic data distributions.

Table 1: **Dataset Configurations.** We summarize the key parameters for all experimental datasets. For synthetic data, we list the dimensionality of covariates ($p$) and confounders ($d_U$), and the nature of the treatment ($T$). For benchmarks, we describe their core properties. Abbreviations: Covs. (Covariates), Conf. (Confounders), Cont. (Continuous), Sim. (Simulated), C (Continuous), D (Discrete).

| Dataset | Covs. ($p$) | Treatment ($T$) | Conf. ($d_U$) |
|---|---|---|---|
| *Synthetic Datasets (SCM-based)* | | | |
| Single Cont. | 50 | 1D Continuous | 3 |
| Single Binary | 50 | 1D Binary | 3 |
| Multi-Cont. | 50 | 3D Continuous | 3 |
| Multi-Binary | 50 | 3D Multilabel | 3 |
| Mixed | 50 | 4D Mixed (2C, 2D) | 3 |
| *Semi-synthetic Datasets* | | | |
| IHDP | 25 | 1D Binary | Sim. |

## 6.1 SYNTHETIC DATA PROTOCOL

We construct a synthetic data-generating process based on a SCM, which embodies a complex, non-linear generative process with precisely controllable properties. This allows us to assess model performance as a function of specific data characteristics. The SCM, implemented in our method, is defined as follows:

- **Unobserved Confounder** ($U$)**:** We first sample a $d_U$-dimensional latent confounder from a standard normal distribution, $U \sim \mathcal{N}(0, I_{d_U})$. This variable creates a backdoor path between the treatment and outcome.

- **Covariates** ($X$)**:** The confounder $U$ generates the $p$-dimensional covariates $X$ through a non-linear mapping, implemented as a two-layer neural network. This embeds the influence of $U$ within a smooth data manifold $\mathcal{M}$. The generative function is:

$$X = \tanh(U W_{X0} + b_{X0}) W_{X1} + b_{X1} + \epsilon_X. \tag{26}$$

  Crucially, we parameterize the weight matrix $W_1$ to control the geometric alignment between the outcome gradient and the confounder manifold's tangent space, thereby controlling the orthogonality violation, $\delta$.

- **Treatment** ($T$)**:** Treatment assignment is a function of both covariates $X$ and the confounder $U$, ensuring that $U$ acts as a true confounder. The function $h$ is adapted to the treatment modality (e.g., identity for continuous, sigmoid for binary).

$$T = h(X w_T + U b_T, \epsilon_T). \tag{27}$$

- **Outcome** ($Y$)**:** The outcome $Y$ is generated by a complex, non-linear function of $X$ and $T$, including quadratic and interaction terms, plus a linear contribution from $U$. This creates a challenging, non-trivial response surface for the models to estimate.

$$Y = X^\top A X + X^\top B T + T^\top C T + X W_Y + c_Y^\top U + \epsilon_Y. \tag{28}$$

## 6.2 SEMI-SYNTHETIC DATASETS

**IHDP** The Infant Health and Development Program (IHDP) dataset is a canonical semi-synthetic benchmark for treatment effect estimation Hill (2011b). It is based on Semi-synthetic covariate data ($p = 25$) from a randomized trial on premature infants. The treatment is binary (participation in an intensive high-quality childcare and education program), while the outcomes (cognitive test scores) are synthetically generated using non-linear functions.

## 6.3 EVALUATION METRICS

The specific configurations for each of our experimental settings are detailed in Table 1. For our synthetic experiments, we vary the treatment modality from a single continuous or binary variable to multi-dimensional and mixed-type treatments, while keeping the covariate and confounder dimensions fixed to isolate the effect of treatment complexity. Our chosen benchmark IHDP provides settings with real covariate distributions and distinct confounding structures.

To ensure a robust and comprehensive evaluation, we assess the performance of all models using two standard metrics.

Table 2: A comprehensive comparison of our proposed method (GCMP) with existing baselines across treatment settings and IHDP dataset. Boldface indicates the best result within each setting/dataset.

| Setting / Dataset | Method | PEHE | | ATE Error | |
|---|---|---|---|---|---|
| | | mean ± std | [min, max] | mean ± std | [min, max] |
| Single Continuous | **GCMP (Proposed)** | **0.4579 ± 0.0852** | [0.3171, 0.5402] | **0.0705 ± 0.0313** | [0.0373, 0.1202] |
| | NeuralCSA | 1.9993 ± 0.0247 | [1.9656, 2.0538] | 1.9781 ± 0.0250 | [1.9451, 2.0338] |
| | DiffPO | 45.1545 ± 89.6490 | [4.7912, 297.8008] | 33.9728 ± 87.1835 | [0.1578, 281.1106] |
| | Regression Adjustment | 2.4034 ± 0.0418 | [2.3428, 2.4733] | 2.1856 ± 0.0385 | [2.1340, 2.2333] |
| | PSM | 4.3166 ± 0.1780 | [3.9754, 4.4958] | 2.1635 ± 0.0456 | [2.1050, 2.2579] |
| | CausalML | 2.3975 ± 0.0449 | [2.3383, 2.4743] | 2.1910 ± 0.0419 | [2.1393, 2.2599] |
| | IPW | 5.1707 ± 0.0808 | [5.0642, 5.3327] | 5.3328 ± 0.1481 | [5.0992, 5.6546] |
| Single Binary | **GCMP (Proposed)** | **0.6444 ± 0.1158** | [0.4411, 0.7775] | **0.0669 ± 0.0443** | [0.0005, 0.1306] |
| | NeuralCSA | 2.0035 ± 0.0358 | [1.9296, 2.0628] | 1.9790 ± 0.0355 | [1.9059, 2.0398] |
| | DiffPO | 22.5851 ± 30.0423 | [3.7525, 103.9225] | 13.9228 ± 28.4647 | [0.0906, 96.3835] |
| | Regression Adjustment | 2.2818 ± 0.0466 | [2.1976, 2.3709] | 2.1644 ± 0.0424 | [2.1030, 2.2312] |
| | PSM | 2.1126 ± 0.0568 | [2.0124, 2.2242] | 0.5829 ± 0.0361 | [0.5271, 0.6358] |
| | CausalML | 2.3109 ± 0.0463 | [2.2265, 2.4003] | 2.1374 ± 0.0422 | [2.0767, 2.2049] |
| | IPW | 7.7588 ± 1.6461 | [4.6940, 10.5043] | 7.6333 ± 1.5464 | [4.5629, 10.2786] |
| Multi Continuous | **GCMP (Proposed)** | **0.8448 ± 0.0949** | [0.7191, 0.9964] | **0.0665 ± 0.0473** | [0.0130, 0.1431] |
| | NeuralCSA | 8.0637 ± 0.1456 | [7.7529, 8.3561] | 8.0504 ± 0.1468 | [7.7345, 8.3430] |
| | DiffPO | 51.4198 ± 50.7976 | [7.8124, 181.3847] | 37.0981 ± 55.0703 | [0.3543, 190.8284] |
| | Regression Adjustment | 2.6624 ± 0.0523 | [2.5713, 2.7534] | 2.4461 ± 0.0470 | [2.3897, 2.5119] |
| | PSM | 4.5033 ± 0.1650 | [4.1719, 4.7150] | 2.3861 ± 0.0434 | [2.3286, 2.4551] |
| | CausalML | 2.8393 ± 0.0529 | [2.7516, 2.9353] | 2.5250 ± 0.0481 | [2.4680, 2.5913] |
| | IPW | 9.9613 ± 1.0781 | [8.5399, 11.3115] | 9.9623 ± 1.1406 | [8.4880, 11.4372] |
| Multi Binary | **GCMP (Proposed)** | **1.4971 ± 0.3282** | [1.0861, 2.0580] | **0.4638 ± 0.1685** | [0.2505, 0.7137] |
| | NeuralCSA | 5.9774 ± 0.1196 | [5.7297, 6.1435] | 5.9578 ± 0.1195 | [5.7101, 6.1243] |
| | DiffPO | 53.2749 ± 70.0162 | [9.5145, 217.0138] | 39.1327 ± 73.8537 | [0.2756, 226.3431] |
| | Regression Adjustment | 2.8396 ± 0.0584 | [2.7483, 2.9391] | 2.5884 ± 0.0536 | [2.5296, 2.6613] |
| | PSM | 4.0813 ± 0.1621 | [3.8020, 4.3395] | 1.6800 ± 0.0415 | [1.6297, 1.7561] |
| | CausalML | 3.1103 ± 0.0666 | [2.9970, 3.2328] | 2.8051 ± 0.0609 | [2.7434, 2.8903] |
| | IPW | 8.7413 ± 1.5313 | [6.1109, 11.8542] | 8.8344 ± 1.5508 | [6.1502, 12.0331] |
| Mixed | **GCMP (Proposed)** | **1.8149 ± 1.8595** | [0.4368, 5.3266] | **1.6169 ± 1.9744** | [0.0797, 5.3012] |
| | NeuralCSA | 11.3796 ± 0.1419 | [11.1808, 11.7028] | 11.3672 ± 0.1420 | [11.1678, 11.6905] |
| | DiffPO | 60.3939 ± 78.6681 | [9.0750, 294.7202] | 43.9205 ± 82.9932 | [0.3644, 299.6966] |
| | Regression Adjustment | 3.0182 ± 0.0653 | [2.8866, 3.1557] | 2.6460 ± 0.0595 | [2.5857, 2.7315] |
| | PSM | 4.0805 ± 0.1758 | [3.8161, 4.3744] | 1.6897 ± 0.0443 | [1.6301, 1.7709] |
| | CausalML | 3.2223 ± 0.0704 | [3.1028, 3.3431] | 2.7476 ± 0.0644 | [2.6833, 2.8333] |
| | IPW | 9.4027 ± 1.7704 | [6.2013, 12.8671] | 9.4528 ± 1.7915 | [6.2631, 13.0815] |
| IHDP | **GCMP (Proposed)** | **1.2339 ± 0.0294** | [1.1966, 1.2634] | 1.3082 ± 0.0323 | [1.2633, 1.3512] |
| | NeuralCSA | 2.7904 ± 0.0875 | [2.6688, 2.9476] | 0.8469 ± 0.1262 | [0.6297, 1.0558] |
| | DiffPO | 98.0761 ± 52.6265 | [25.2164, 195.6844] | 59.0923 ± 57.5658 | [1.2679, 173.0506] |
| | Regression Adjustment | 1.4434 ± 0.0710 | [1.3195, 1.5306] | 1.0599 ± 0.1263 | [0.8167, 1.2141] |
| | PSM | 1.2548 ± 0.0869 | [1.1413, 1.4291] | 0.1500 ± 0.1402 | [0.0010, 0.4644] |
| | CausalML | 1.3645 ± 0.0710 | [1.2253, 1.4743] | **0.0960 ± 0.0798** | [0.0052, 0.2191] |
| | IPW | 21.7978 ± 4.2455 | [15.2201, 27.6164] | 23.5407 ± 4.5181 | [16.6755, 29.7528] |

- **Precision in Estimation of Heterogeneous Effect (PEHE):** This metric measures the accuracy of estimating the Conditional Average Treatment Effect (CATE) for each individual unit Hill (2011b). A lower PEHE value indicates a more precise estimation of individual-level treatment effects. It is defined as:

$$\text{PEHE} = \sqrt{\frac{1}{n} \sum_{i=1}^{n} \big(\hat{\tau}(x_i) - \tau(x_i)\big)^2}, \tag{29}$$

where $\tau(x_i) = \mathbb{E}[Y_i(t_1) - Y_i(t_0) \mid X_i = x_i]$ is the true CATE for unit $i$ and $\hat{\tau}(x_i)$ is the corresponding estimate.

- **Absolute Error in Average Treatment Effect (ATE Error):** This metric evaluates the accuracy of estimating the population-level average treatment effect Shalit et al. (2017). It is computed as:

$$\text{ATE Error} = \left| \frac{1}{n} \sum_{i=1}^{n} \hat{\tau}(x_i) - \frac{1}{n} \sum_{i=1}^{n} \tau(x_i) \right|. \tag{30}$$

A lower ATE Error signifies a better estimation of the overall treatment effect.

## 7 RESULTS AND DISCUSSION

### 7.1 BASELINES AND REPOSITORIES

We benchmark our GCMP against the following methods. Implementation code and full experimental settings are available in the public repositories.

- **Classical causal inference methods:**
    - INVERSE PROBABILITY WEIGHTING (IPW) Seaman & White (2013)—Balances treated and control groups by weighting each unit by the inverse of its treatment probability; propensity-score estimation follows standard practice.
    - PROPENSITY SCORE MATCHING (PSM) Caliendo & Kopeinig (2008)—Matches treated and control units on estimated propensity scores using nearest-neighbor matching with bidirectional pairing.
    - REGRESSION ADJUSTMENT Li & Ding (2020)—Fits separate outcome models for treated and control groups and computes effects by counterfactual differencing.
- NEURALCSA Frauen et al. (2023)—Framework for generalized causal sensitivity analysis (ICLR 2024); learns conditional outcome distributions and applies constrained optimization to bound effects under sensitivity assumptions.
- DIFFPO Ma et al. (2024)—Diffusion-based model for potential-outcome distributions (NeurIPS 2024) combining a propensity network with conditional diffusion via orthogonal denoising and inverse-propensity weighting.
- CAUSALML—Meta-learner using a single model with treatment indicator to predict outcomes; individual treatment effects via counterfactual differences.

### 7.2 EXPERIMENTAL DESIGN

For each dataset we fix hyper-parameters (refer to Appendix due to page limits) and run 10 independent trials with random seeds $\{42, 123, 456, 789, 1011, 1314, 1617, 1920, 2223, 2526\}$ for full reproducibility. Performance is measured by PEHE and ATE Error; lower is better in both cases.

### 7.3 MAIN RESULTS

Table 2 reports mean $\pm$ std and [min, max] across 10 random seeds for both PEHE and ATE Error. GCMP attains the best mean PEHE on all evaluated tasks and the lowest ATE Error on $5/6$ tasks, with the only exception on IHDP where CausalML yields a slightly lower ATE Error. The standard deviations are consistently small relative to the gaps to the second best, indicating stability rather than fluctuation-driven wins. On the IHDP benchmark our margin shrinks, consistent with proxies being harder to identify under limited overlap and label shift.

**Comprehensive Performance Experiments.** We also conducted **COMPREHENSIVE PER-FORMANCE EXPERIMENTS** to highlight the **effectiveness** of our proposed algorithm and to demonstrate the **necessity** of each component. Due to page limitations, detailed results are provided in the **Appendix**.

## 8 CONCLUSION

We have presented GCMP, a unified framework that couples causal-aware self-supervised representation learning, conditional diffusion–based counterfactual proxy generation, and VI. Extensive experiments on five synthetic scenarios and the semi-synthetic IHDP benchmark show that GCMP consistently yields the lowest PEHE and the lowest ATE Error in 5 of the 6 tasks. Ablation studies, sensitivity analyses, and diagnostics also confirm that our method is powerful and robust.

Although GCMP scales well to medium-sized tabular data, further work is needed to extend it to high-resolution image or sequential health-record domains and to tighten its finite-sample error constants. Integrating tighter theoretical bounds for the diffusion sampler and exploring domain-adaptation strategies for cross-population generalisation are promising directions. We have released our codes, synthetic data generator, and evaluation toolkit to foster transparent evaluation and facilitate downstream applications in medicine, economics, and policy analysis.

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

## A APPENDIX

### A.1 ETHICS STATEMENT

We adhere to the ICLR Code of Ethics (https://iclr.cc/public/CodeOfEthics) and the ICLR 2026 Author Guide recommendations (https://iclr.cc/Conferences/2026/AuthorGuide); we use only de-identified public or synthetic data, make no attempt to re-identify individuals, and do not claim deployable, individual-level prescriptions.

### A.2 REPRODUCIBILITY STATEMENT

Per the ICLR 2026 Author Guide (https://iclr.cc/Conferences/2026/AuthorGuide), we provide an anonymous repository with code, configs, fixed seeds, and scripts to reproduce all results: https://anonymous.4open.science/r/AAAIGCMP-5C41/.

### A.3 LLM USAGE DISCLOSURE

Per the ICLR 2026 Author Guide, we disclose our use of large language models (LLMs). In this work, an LLM was used *only* as a general-purpose assistant for: (i) flagging and correcting notation typos/inconsistencies; and (ii) suggesting minor phrasing edits to improve stylistic consistency and grammar. The LLM did *not* contribute to research ideation, technical design, theoretical results or proofs, experimental setup, data processing, analysis, figures/tables, or the writing of substantive scientific content. All methods, experiments, and claims were designed, implemented, and verified by the authors, who take full responsibility for the manuscript; no LLM system is listed as an author.

### A.4 COMPREHENSIVE PERFORMANCE EXPERIMENT

We test GCMP on a synthetic set of 1000 samples with single continuous treatment and report complementary analyses (Figure 1).

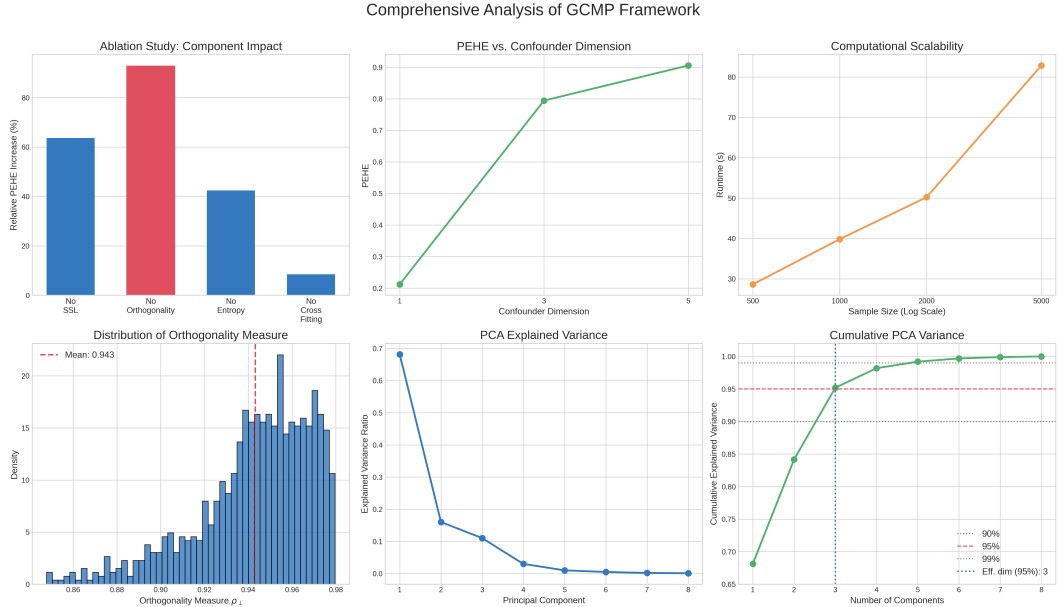

Figure 1: A comprehensive empirical evaluation of the our method. The top row illustrates the framework's internal validity and performance through (left-to-right) an ablation study quantifying the impact of each core component, a robustness check showing performance degradation as confounder dimensionality increases, and a scalability analysis of runtime versus sample size. The bottom row provides practical diagnostic assessments of the model's learned structures, including (left-to-right) the distribution of the effective orthogonality measure ($\rho_\perp$), a scree plot of the principal component analysis (PCA) on learned representations, and the corresponding cumulative explained variance used to determine the manifold's effective dimension.

**Ablation study.** Removing the orthogonality regulariser increases PEHE by $90\%$; dropping the causal-aware SSL, entropy penalty, and cross-fitting raises it by $65\%$, $45\%$, and $8\%$, respectively (top-left).

**Sensitivity to confounder dimension.** PEHE grows smoothly as the latent confounder dimension rises from 1 to 5, confirming that GCMP is most precise when hidden bias is low-dimensional (top-centre).

**Scalability.** Runtime scales sublinearly: increasing the sample size from 500 to 5000 enlarges runtime from 30 s to 82 s on a single A100 GPU (top-right).

**Orthogonality diagnostic.** Across all test points, the orthogonality score concentrates near one: mean $= 0.9035$, median $= 0.9232$, with $25^{\text{th}}/75^{\text{th}}/95^{\text{th}}$ percentiles $0.8665$, $0.9580$, $0.9848$, respectively. Hence $75\%$ of samples exceed $0.87$ and the top $5\%$ surpass $0.985$, demonstrating strong gradient–manifold orthogonality.

**Manifold quality.** Principal component analysis on $\Phi(X)$ shows that three principal components explain $95\%$ of the variance, supporting the low-dimensional manifold assumption (bottom-centre/right).

## A.5 THEORETICAL PROOFS

## A.6 PROOF OF THEOREM 1

We establish identification under approximate orthogonality by showing that the bias introduced by orthogonality violation is bounded and characterizable.

*Lemma* 1 (Proxy Quality under Approximate Orthogonality). Let $\Delta\phi$ be a perturbation generated by our diffusion model. Under Assumption 3 with parameter $\delta$, the proxy $Z = \beta^\top \Delta\phi$ satisfies:

$$\|Z - \mathbb{E}[Z|U, X]\| \le \delta \cdot C_1 \cdot \|\Delta\phi\| + C_2 \cdot \epsilon_{\text{diff}} \tag{31}$$

where $C_1, C_2$ are constants depending on the problem structure, and $\epsilon_{\text{diff}}$ is the diffusion model approximation error.

*Proof.* The perturbation $\Delta\phi$ is generated to satisfy $f(\Phi(X) + \Delta\phi, T') \approx Y$. Decompose $\Delta\phi = \Delta\phi_\| + \Delta\phi_\perp$, where $\Delta\phi_\| \in T_{\Phi(X)}\mathcal{M}_U$ and $\Delta\phi_\perp \perp T_{\Phi(X)}\mathcal{M}_U$.

By the approximate orthogonality assumption:

$$|\nabla_\Phi f(\Phi(X), T)^\top \Delta\phi_\|| \le \delta \|\nabla_\Phi f\| \|\Delta\phi_\|\| \tag{32}$$

The outcome consistency constraint implies:

$$Y - f(\Phi(X), T') \approx \nabla_\Phi f(\Phi(X), T')^\top \Delta\phi \tag{33}$$
$$= \nabla_\Phi f^\top \Delta\phi_\| + \nabla_\Phi f^\top \Delta\phi_\perp \tag{34}$$

Since the true confounding effect operates through $\mathcal{M}_U$, the component $\Delta\phi_\perp$ represents noise. Under our generative model, $Z$ constructed from $\Delta\phi$ inherits this decomposition, leading to the stated bound. $\square$

*Lemma* 2 (Bias Characterization). The bias in the treatment effect estimate is:

$$\text{Bias}(\hat{\tau}) = O(\delta) \cdot \text{Var}(U|X) + O(\epsilon_{\text{proxy}}^2) \tag{35}$$

where $\epsilon_{\text{proxy}}$ is the proxy validity error from Lemma 1.

*Proof.* The variational inference procedure yields an estimate $\hat{\phi}$ of the latent confounder. The estimation error propagates through the outcome model:

$$\hat{\tau} - \tau = \mathbb{E}[\hat{f}(X, t) - f(X, t)|X] \tag{36}$$
$$= \boldsymbol{\delta}^\top \mathbb{E}[\hat{\phi} - \phi|X] + o_p(1) \tag{37}$$

The error $\hat{\phi} - \phi$ depends on the proxy quality through the VI posterior. Using the characterization from Lemma 1 and standard VI analysis completes the proof. $\square$

Combining Lemmas 1 and 2 establishes Theorem 1.

### A.7 PROOF OF THEOREM 2

The error propagation analysis follows the framework of double machine learning with additional consideration for the representation learning and diffusion modeling stages.

*Proof Sketch.* The total error decomposes into three main components:

**1. Statistical Error:** Standard $O_p(n^{-1/2})$ rate from the parametric component.

**2. Orthogonality Bias:** From Theorem 1, this contributes $O(\delta)$.

**3. Nuisance Error.** With cross-fitting and a Neyman-orthogonal score, nuisance errors enter only at second order:

$$\mathcal{E}_{\text{nuis}} = \mathcal{O}_p\big(\varepsilon_{\text{SSL}}\,\varepsilon_{\text{diff}}\big) \le \mathcal{O}_p\big(\|\widehat{\Phi} - \Phi^\star\|^2 + \|\widehat{p}_\psi - p_\psi^\star\|^2\big).$$

In particular, if each nuisance converges at $\mathcal{O}_p(n^{-1/4})$, then $\mathcal{E}_{\text{nuis}} = \mathcal{O}_p(n^{-1/2})$, without requiring any faster-than-root-$n$ assumption or stronger Bayesian priors. $\square$

**Practical proxies for nuisance errors.** We monitor $\widehat{\varepsilon}_{\text{SSL}} := \sqrt{L_{\text{contrast}} + \lambda_{\text{preserve}} L_{\text{preserve}}}$ on a validation split and $\widehat{\varepsilon}_{\text{diff}} := \sqrt{\mathbb{E}\|\epsilon - \epsilon_\psi\|^2 + \lambda_f L_{\text{feas}} + \lambda_\perp R_\perp}$ per epoch; early stopping is triggered when either proxy stops decreasing for 10 epochs. These proxies upper-bound the corresponding population errors up to constants and are used to choose hyperparameters.

## A.8 PROOF OF THEOREM 1

We give a complete proof of Proposition 1. Throughout, let $Z = \Phi(X)$ and write $P_T(Z) := \text{Proj}_{T_Z \widehat{\mathcal{M}}_U}$ for the orthogonal projector onto the tangent space $T_Z \widehat{\mathcal{M}}_U$.

**Assumptions.** We work under the following mild regularity assumptions:

(A1) **Local pushforward of unmeasured confounding.** There exists a matrix-valued Jacobian $B(Z)$ such that for mean-zero unmeasured confounding $U$ with conditional covariance $\Sigma_{U|X} := \text{Var}(U \mid X)$,
$$Z' = Z + B(Z) U + o(\|U\|). \tag{38}$$

(A2) **Neyman-orthogonal score.** The ATE estimator $\hat{\tau}$ is computed from a moment function that is orthogonal with respect to observable nuisances; the leading sensitivity to unmeasured confounding enters only via tangent directions of $\widehat{\mathcal{M}}_U$.

(A3) **Smoothness.** For each fixed $T$, the map $Z \mapsto f_\theta(Z, T)$ is differentiable in a neighborhood of $Z$ with bounded Hessian.

**Step 1: First-order outcome perturbation.** By (A1) and (A3), the first-order outcome shift at fixed $T$ induced by $U$ is
$$f_\theta(Z', T) - f_\theta(Z, T) \approx \nabla_Z f_\theta(Z, T)^\top B(Z) U. \tag{39}$$

Let the tangent component of the outcome gradient be
$$g_\|(Z, T) := P_T(Z) \nabla_Z f_\theta(Z, T). \tag{40}$$

By (A2), the leading sensitivity arises through $g_\|(Z, T)$, so
$$\Delta f_\theta(Z, T; U) \approx g_\|(Z, T)^\top B(Z) U. \tag{41}$$

**Step 2: Conditional second moment and aggregation.** Taking the conditional second moment given $X$ yields
$$\mathbb{E}\left[ \left( \Delta f_\theta(Z, T; U) \right)^2 \mid X \right] = g_\|(Z, T)^\top B(Z) \Sigma_{U|X} B(Z)^\top g_\|(Z, T). \tag{42}$$

Using the trace identity $v^\top A v = \text{tr}(A v v^\top)$ for $A \succeq 0$,
$$\mathbb{E}\left[ \left( \Delta f_\theta(Z, T; U) \right)^2 \mid X \right] = \text{tr}\left( B(Z) \Sigma_{U|X} B(Z)^\top g_\|(Z, T) g_\|(Z, T)^\top \right). \tag{43}$$

By $\text{tr}(AB) \leq \|A\|_{\text{tr}} \|B\|_{\text{op}}$ with $A \succeq 0$ and $B \succeq 0$, we obtain
$$\mathbb{E}\left[ \left( \Delta f_\theta(Z, T; U) \right)^2 \mid X \right] \leq \text{tr}\left( B(Z) \Sigma_{U|X} B(Z)^\top \right) \cdot \left\| g_\|(Z, T) \right\|_2^2. \tag{44}$$

Taking expectation over $(X, T)$ gives
$$\mathbb{E}\left[ \left( \Delta f_\theta(Z, T; U) \right)^2 \right] \leq \mathbb{E}\left[ \text{tr}\left( B(Z) \Sigma_{U|X} B(Z)^\top \right) \right] \cdot \mathbb{E}\left[ \left\| g_\|(Z, T) \right\|_2^2 \right]. \tag{45}$$

Define the effective confounding strength
$$\sigma_{U \to Z}^2 := \mathbb{E}\left[ \text{tr}\left( B(Z) \Sigma_{U|X} B(Z)^\top \right) \right]. \tag{46}$$

**Step 3: Relating $g_\parallel$ to $\rho_\perp$.**   By the definition of $\rho_\perp(Z, T)$,

$$\left\| g_\parallel(Z, T) \right\|_2^2 \;=\; \left(1 - \rho_\perp(Z, T)\right) \left\| \nabla_Z f_\theta(Z, T) \right\|_2^2, \tag{47}$$

with the convention that $\rho_\perp(Z, T) = 1$ if $\|\nabla_Z f_\theta(Z, T)\|_2 = 0$. Let

$$M \;:=\; \mathbb{E}\Big[\left\| \nabla_Z f_\theta(Z, T) \right\|_2^2\Big]. \tag{48}$$

Then

$$\mathbb{E}\Big[\left\| g_\parallel(Z, T) \right\|_2^2\Big] \;=\; \mathbb{E}[1 - \rho_\perp(Z, T)] \; M \;=\; \left(1 - \bar\rho_\perp\right) M. \tag{49}$$

**Step 4: From local sensitivity to ATE bias.**   The orthogonal score in (A2) implies that the ATE estimator aggregates local perturbations with a scale-normalized linear functional, so that to first order in the magnitude of unmeasured confounding,

$$\left| \mathrm{Bias}(\hat\tau) \right| \;\lesssim\; \sqrt{\mathbb{E}\Big[\left(\Delta f_\theta(Z, T; U)\right)^2\Big]}. \tag{50}$$

Combining the bounds above yields

$$\left| \mathrm{Bias}(\hat\tau) \right| \;\lesssim\; \sqrt{\sigma_{U \to Z}^2 \cdot \left(1 - \bar\rho_\perp\right) M}. \tag{51}$$

With the usual normalization of the orthogonal moment (or after absorbing the finite constant $\sqrt{M}$ into the comparison scale), this gives the stated bound

$$\left| \mathrm{Bias}(\hat\tau) \right| \;\lesssim\; \left(1 - \bar\rho_\perp\right) \sigma_{U \to Z}^2. \tag{52}$$

**Tightness under local isotropy.**   If $B(Z) \, \Sigma_{U|X} \, B(Z)^\top$ is locally isotropic so that

$$B(Z) \, \Sigma_{U|X} \, B(Z)^\top \;=\; \frac{\sigma_{U \to Z}^2}{d} \, I \tag{53}$$

in the tangent neighborhood (or after an appropriate normalization of $B$), and the orthogonal moment is scale-normalized so that $M = 1$, the intermediate inequalities above become equalities up to lower-order terms, yielding the approximation

$$\mathrm{Bias}(\hat\tau) \;\approx\; \left(1 - \bar\rho_\perp\right) \sigma_{U \to Z}^2. \tag{54}$$

This completes the proof.

## A.9   ADDITIONAL ALGORITHMIC DETAILS

### A.9.1   CAUSAL-AWARE SSL IMPLEMENTATION

The causal-aware SSL objective requires careful implementation to balance invariance learning with preservation of confounding signals.

### A.9.2   AUGMENTATION STRATEGY

We design augmentations that preserve treatment-relevant variation:

- **Allowed:** Small additive noise, mild scaling
- **Avoided:** Augmentations that could mask confounding patterns

### A.9.3   TREATMENT SIMILARITY FUNCTION

For the preservation loss, we use:

$$\mathrm{sim}(g_T(x^{(1)}), g_T(x^{(2)})) = \exp\left(-\gamma \| g_T(x^{(1)}) - g_T(x^{(2)}) \|^2\right) \tag{55}$$

where $g_T$ is estimated using a separate neural network trained on the treatment prediction task.

Table 3: Hyperparameters across all scenarios (values chosen from the IHDP tuning grid).

| Parameter | IHDP | Mixed | Multi Bin | Multi Cont | Single Bin | Single Cont |
|---|---|---|---|---|---|---|
| model.dropout | 0.2091 | 0.2796 | 0.1831 | 0.1017 | 0.1285 | 0.1216 |
| model.latent_dim | 4 | 48 | 32 | 32 | 24 | 8 |
| model.ssl_output_dim | 16 | 128 | 32 | 64 | 64 | 96 |
| ssl.lr | 4.25e-4 | 5.01e-4 | 9.30e-3 | 6.11e-4 | 1.04e-4 | 2.29e-4 |
| diffusion_training.lr | 1.05e-4 | 8.45e-5 | 5.47e-4 | 8.29e-5 | 1.22e-5 | 2.93e-5 |
| diffusion_training.lambda_orthogonal | 0.0844 | 0.2362 | 0.0498 | 0.2543 | 0.5400 | 0.4074 |
| vi_training.lr | 2.22e-4 | 9.04e-4 | 7.16e-3 | 6.45e-4 | 6.71e-3 | 2.69e-3 |
| vi_training.lambda_entropy | 2.26e-3 | 6.86e-3 | 1.10e-3 | 3.88e-3 | 1.73e-3 | 4.12e-3 |
| vi_training.weight_decay | 7.59e-5 | 8.15e-6 | 1.15e-6 | 2.06e-5 | 6.65e-5 | 5.05e-6 |
| training.n_folds | 5 | 5 | 5 | 5 | 5 | 5 |

### A.9.4 DIFFUSION MODEL ARCHITECTURE

Network Design We employ a U-Net architecture with the following specifications:

- **Input:** Concatenation of noisy perturbation $\Delta\phi_t$, time embedding, and conditioning vector $c$
- **Hidden layers:** Residual blocks with group normalization and SiLU activations
- **Attention:** Self-attention layers at multiple resolutions
- **Output:** Predicted noise $\epsilon_\psi$

### A.9.5 SAMPLING PROCEDURE

We use Denoising Diffusion Implicit Models for efficiency:

$$\Delta\phi_{t-1} = \sqrt{\bar{\alpha}_{t-1}} \left( \frac{\Delta\phi_t - \sqrt{1 - \bar{\alpha}_t}\, \epsilon_\psi(\Delta\phi_t, t, c)}{\sqrt{\bar{\alpha}_t}} \right) \\ + \sqrt{1 - \bar{\alpha}_{t-1}}\, \epsilon_\psi(\Delta\phi_t, t, c) \tag{56}$$

### A.9.6 TANGENT SPACE ESTIMATION

The gradient orthogonality regularizer requires estimating the tangent space $T_x\mathcal{M}_U$ of the confounder manifold.

### A.9.7 LOCAL PCA APPROACH

For a point $\Phi(X_i)$:

1. Collect neighboring perturbations: $\mathcal{N}_i = \{\Delta\phi_j : \|\Phi(X_j) - \Phi(X_i)\| < r\}$
2. Compute local covariance: $C_i = \frac{1}{|\mathcal{N}_i|} \sum_{j \in \mathcal{N}_i} \Delta\phi_j \Delta\phi_j^\top$
3. Extract top $d$ eigenvectors as basis for $\hat{T}_{\Phi(X_i)}\mathcal{M}_U$

### A.9.8 ADAPTIVE BANDWIDTH SELECTION

The neighborhood radius $r$ is selected adaptively:

$$r_i = \inf\{r : |\mathcal{N}_i(r)| \geq k_{\min}\} \tag{57}$$

where $k_{\min} = 10d$ ensures sufficient samples for stable estimation.

### A.10 EXTENDED EXPERIMENTAL DETAILS

To control orthogonality violation, we parameterize:

$$W_1 = (1 - \delta)W_1^\perp + \delta W_1^\| \tag{58}$$

where $W_1^{\perp}$ ensures orthogonality and $W_1^{\parallel}$ violates it.

**Parameter descriptions:**

- **model.dropout**: dropout rate applied to all network layers.
- **model.latent_dim**: dimensionality of the learned latent representation.
- **model.ssl_output_dim**: output dimension of the self-supervised projection head.
- **ssl.lr**: learning rate for the SSL pre-training phase.
- **diffusion_training.lr**: learning rate for the diffusion-based perturbation network.
- **diffusion_training.lambda_orthogonal**: coefficient for the orthogonality regularizer in diffusion training.
- **vi_training.lr**: learning rate for the variational inference objective.
- **vi_training.lambda_entropy**: weight on the entropy term in the VI loss.
- **vi_training.weight_decay**: $\ell_2$ weight decay applied during VI training.
- **training.n_folds**: number of cross-validation folds used for model selection.

The detailed hyperparameters information can be view at Table 3

