# OpenReview forum: "Generative Counterfactual Manifold Perturbation: A Robust Framework for Treatment Effect Estimation with Unobserved Confounders"
_ICLR.cc/2026/Conference — ICLR 2026 Conference Desk Rejected Submission_

### Official Review · Reviewer_PxTe · 2025-10-28

**Soundness:** 2
**Presentation:** 2
**Contribution:** 2
**Rating:** 2
**Confidence:** 3

**Summary:**

This paper addresses the challenge of hidden confounding in causal inference by proposing a novel framework: Generative Counterfactual Manifold Perturbation (GCMP). GCMP unifies causal-aware self-supervised learning, conditional diffusion-based counterfactual proxy generation, and adaptive variational inference. The authors conduct a series of experiments to empirically validate the effectiveness of their approach.

**Strengths:**

1. The theoretical derivations are solid and provide strong support for the proposed method.
2. The focus on causal inference aligns well with the interests of the ICLR community.
3. The experimental results are clearly presented and substantiate the claims made in the paper.

**Weaknesses:**

1. The overall workflow of the proposed approach is not clearly described, making it challenging to fully understand the method. Including a schematic diagram or pseudo-code would enhance clarity.

2. The related works section lacks coverage of classical causal inference methods, particularly recent developments within the last three years.

3. The definition and details of the manifolds used (e.g., simplex manifold, Stiefel manifold) are unclear.

4. The motivation for introducing local PCA is unclear. Is it intended to serve as a retraction operator or for some other specific purpose? Please clarify the rationale and theoretical justification.

5. Why is the DDPM framework specifically chosen for the generative model? What would be the effect of using a variance explosion SDE framework instead [1]? A comparison or justification would strengthen the paper.

6. The explicit form of the similarity function $\text{Sim}$ in Equation (10) is not provided. Please clarify its definition.

7. While variational inference can approximate the posterior, the approach appears to be a form of amortized variational inference, which may introduce an amortization gap. What would be the impact of using Monte Carlo-based approaches instead?

---
References:
[1]. Score-Based Generative Modeling through Stochastic Differential Equations

**Questions:**

Please see the weaknesses.

---

### Official Review · Reviewer_37Rv · 2025-10-30

**Soundness:** 3
**Presentation:** 2
**Contribution:** 2
**Rating:** 4
**Confidence:** 4

**Summary:**

To address the challenge of unobserved confounders that induce spurious associations, the authors propose a framework named Generative Counterfactual Manifold Perturbation (GCMP), which integrates causal-aware self-supervised learning, conditional diffusion-based counterfactual proxy generation, and adaptive variational inference. Experimental results demonstrate that GCMP consistently outperforms state-of-the-art methods.

**Strengths:**

Capable of handling diverse treatment types within a unified framework, including continuous, binary, and multi-dimensional treatments.
The experiments cover multi-dataset comparison and ablation studies, which effectively verifies the effectiveness and robustness of the method.

**Weaknesses:**

1. The paper shows limited novelty, as the methodology combines three existing modules without sufficiently clarifying the specific challenges each addresses and how they interconnect functionally.

2. It is strongly recommended that the authors include an overall framework diagram.

3. The authors extensively cite prior work but does not discuss alternatives to normalizing flows for counterfactual, such as energy-based models or manifold-based generative approaches.

4. The authors report experimental results, but the evaluation misses several strong baselines (e.g., TarNet, CFRNet, DragonNet, CEVAE and GANITE), which are essential.

5. The code link provided appears to be incorrect and I could not access the related code.

**Questions:**

As noted in the Weaknesses.

---

### Official Review · Reviewer_nusj · 2025-10-30

**Soundness:** 3
**Presentation:** 1
**Contribution:** 2
**Rating:** 2
**Confidence:** 3

**Summary:**

This paper introduces a new framework for estimating CATE, where a new loss was proposed and several existing architectures were combined. Theoretical analysis and empirical validation were presented.

**Strengths:**

The paper addresses causal inference with unobserved confounders, which is a critical field of study.

The codebase is well-maintained.

**Weaknesses:**

Unverified assumption: this paper does not seem to rely on the standard causal inference assumptions (e.g., consistency, exchangeability, etc.) and requires three new assumptions that the authors have introduced. While the authors claimed that the assumptions are easier to verify empirically, no experimental results were shown to validate these assumptions, and no verbal explanations were provided to help the audience interpret the assumptions.

Presentation clarity: The proposed method contains many modules. However, no summarizing statement of how the three modules interact with each other and are integrated into the framework. Therefore, the overall arrangement of the paper is chaotic and hard to understand, and it is difficult for me to appreciate the novelty of its architectural design.

This also applies to the theory section of the paper: the proved theoretical results are not explained for their role in the framework and contribution to the overall literature. Therefore, it is also unclear to me the significance of these results to the overall quality of the paper.

Redundant information in the main paper: some details in the paper are redundant and can be moved to the appendix. For example, the definition of PHEH is relatively standard, and its expression does not need to be shown in the main paper. After saving these spaces, the omitted experiment results could be moved to the main paper.

Weak experiments: there are a lot of aspects of the proposed method unverified, which could benefit from further ablation studies.

Misleading wording: Line 232, the use of "introduce" is misleading, as it typically represents that what is about to be discussed is novel, while what is shown in the paper is generic, so it seems like a potential overclaim. I suggest replacing "introduce" with "construct".

**Questions:**

Could the authors please provide illustrations or provide an overview of how the components in the framework are assembled?

Could the authors present ablation studies on how each component in the proposed method contributes to the final performance?

Could the authors provide comparison results between your proposed method and proximal causal inference baselines?

---

### Note · Program_Chairs · 2026-01-17
**Submission Desk Rejected by Program Chairs**

The following references in this submission do not refer to real documents and/or have major errors in bibliographic information:

 Peter M. Spieth, Emily Palmer, and Miguel Santos. Randomised controlled trials-why they often fall short and what observational studies offer. Annals of Medicine, 52(11):523-532, 2020.
Kun Zhang and Bernhard Schölkopf. Identifiability in causal representation learning. Foundations and Trends in Machine Learning, 2023.